# How stra(i)nge are your controls? A comparative analysis of metabolic phenotypes in commonly used C57BL/6 substrains

**Annesha Sil**[ID]**‡\***, **Marina Souza Matos‡**, **Mirela Delibegovic**[ID], **Bettina Platt\***

Institute of Medical Sciences, School of Medicine, Medical Sciences & Nutrition, Foresterhill, University of Aberdeen, Aberdeen, Scotland, United Kingdom

‡ AS and MSM contributed equally to this work and are joint first authors.
\* b.platt@abdn.ac.uk (BP); annesha.sil@abdn.ac.uk (AS)

## Abstract

In recent years, insufficiently characterised controls have been a contributing factor to irre-producibility in biomedical research including neuroscience and metabolism. There is now a growing awareness of phenotypic differences between the C57BL/6 substrains which are commonly used as control animals. We here investigated baseline metabolic characteristics such as glucose regulation, fasted serum insulin levels and hepatic insulin signalling in five different C57BL/6 substrains (N, J, JOla, JRcc) of both sexes, obtained from two commercial vendors, Charles River Laboratories (Crl) and Envigo (Env). Our results indicate systematic and tissue-specific differences between substrains, affected by both vendor and sex, in all parameters investigated, and not necessarily mediated by the presence of the $Nnt^{C57BL/6J}$ mutation. Not only were there differences between 6J and 6N as expected, all three 6J substrains exhibited different profiles, even from the same breeder. Two distinct metabolic profiles were identified, one in which low insulin levels resulted in impaired glucose clearance (6JCrl; both sexes) and the other, where sustained elevations in fasted basal insulin levels led to glucose intolerance (male 6JRccEnv). Further, 6JRccEnv displayed sex differences in both glucose clearance and hepatic insulin signalling markers. In comparison, the two 6N substrains of either sex, irrespective of vendor, did not exhibit considerable differences, with 6NCrl animals presenting a good choice as a healthy baseline 'control' for many types of experiments. Overall, our data emphasise the importance of selecting and characterising control subjects regarding background, sex, and supplier to ensure proper experimental outcomes in biomedical research.

## Introduction

In recent years, biomedical research has found itself at the centre of a reproducibility crisis [1]. Among the many issues identified, one crucial factor is the use of insufficiently characterised controls, especially in studies involving the use of animals, which can distort experimental

**Data Availability Statement:** All relevant data are within the paper and its Supporting Information files. All raw data pertaining to this paper can be

found in the Supporting Information files and are stored according to Institutional guidelines.

**Funding:** The authors received no specific funding for this work.

**Competing interests:** The authors have declared that no competing interests exist.

outcomes and conclusions drawn. Concurrently, there has been increasing awareness of the differences between inbred mouse substrains used in different areas of biomedical research, including neuroscience and metabolism research [2]. Here, 'inbred' strains are defined as any strain where brother-sister mating has occurred for at least 20 consecutive generations, whereas 'substrain' is a genetically distinct branch that has significantly deviated from the original founding strain. One of the most commonly used inbred mouse strains in biomedical research are the C57BL/6 lines, which can be purchased from multiple commercial vendors around the globe, including Charles River (Crl), Envigo (Env), and The Jackson Laboratory (JAX) [3]. The C57BL/6J (6J) and C57BL/6N (6N) substrains are usually the primary choice for many types of biomedical studies due to ease of breeding and background stability, which also make these a good choice for genetic modifications [4]. Literature on these 'wild-type' strains often do not take into consideration the genetic make-up and phenotypic differences already reported [5–7] and can be plagued by insufficient information provided, offering only "C57" or "C57BL6" labels without further details.

Several single nucleotide polymorphisms (SNPs) have been identified between 6J and 6N mice [8]. Regulatory and coding regions containing SNPs can severely change gene expression and function [6]. In 2005, a spontaneous in-frame five-exon deletion in the *Nnt* gene was identified in 6J mice [5]. The affected nicotinamide nucleotide transhydrogenase (NNT) protein is a mitochondrial enzyme responsible for the reduction of $NADP^+$ to NADPH, NADPH is required for biosynthesis and redox homeostasis. Consequently, the *Nnt* mutation has been linked to metabolic phenotypes observed in 6J mice, including impaired insulin secretion and glucose intolerance [5, 9–11].

At present, conflicting data exist regarding metabolic differences between 6J and 6N mice, including glucose tolerance and body weight [4]. 6J and 6N animals develop glucose intolerance when on high-fat diet (HFD), yet 6J mice showed higher glucose levels [12] and lower insulin secretion than 6N mice [13]. However, other studies have reported no major metabolic differences between 6J and 6N [4, 14, 15]. Hence, *Nnt* is unlikely to be the sole contributing factor.

Regarding phenotypic differences between the 6J substrains from different suppliers, SNP analysis showed that 6JCrl, 6JArc, and 6J (JAX) were genetically identical, but those from other suppliers displayed substrain-specific polymorphisms beyond *Nnt* [16]. For example, 6JOlaHsd (Envigo) mice have a loss-of-function deletion in the alpha-synuclein (*Scna*) and multimerin 1 (*Mmrn1*) genes. Remarkably, all 6N substrains have similar genetic profiles independent of the supplier [17], but display a nonsense mutation in the *Crb1* gene that promotes retinal degeneration 8 (*rd8* mutation) [4]. Table 1, adapted from the Envigo substrain information, lists common genetic mutations in the substrains utilised in our study [18].

**Table 1. List of different substrains utilised with some important known genetic deletions.**

| Substrain | Supplier | Genetic deletions | | | |
|---|---|---|---|---|---|
| | | *Nnt*[a] | *Scna*[b] | *Mnrn1*[c] | *Rd8*[d] |
| 6JCrl | Charles River Laboratories | Yes | No | No | No |
| 6JOlaEnv | Envigo | No | Yes | Yes | No |
| 6JRccEnv | Envigo | No | No | No | No |
| 6NCrl | Charles River Laboratories | No | No | No | Yes |
| 6NEnv | Envigo | No | No | No | Yes |

[a]*Nnt* = nicotinamide nucleotide transhydrogenase; encodes an integral inner mitochondrial membrane protein linked to metabolic alterations; [b]*Scna* = alpha-synuclein; abnormal alpha-synuclein accumulation implicated in Parkinson's and other synucleinopathies; [c]*Mnrn1* = multimerin-1; stored platelet and endothelial cell adhesive protein; [d]*Rd8* = retinal degeneration-8; mutations linked to macular degeneration and other age-related vision loss which can complicate ophthalmological studies/behavioural testing [18].

So far, hepatic insulin signalling central for systemic glucose homeostasis has not been characterised adequately in C57 substrains. Altered insulin signalling can ultimately lead to insulin resistance, hyperglycaemia, and type 2 diabetes (T2D). Insulin receptor (IR) activation triggers phosphorylation of protein kinase B (AKT) via phosphoinositide-3-phosphate kinase (PI3K). AKT can regulate multiple pathways to regulate glucose and lipid homeostasis, e.g., glycogen synthesis and inhibition of the glycogen synthase kinase 3β (GSK3β). Additionally, activated AKT can lead to an increase in mammalian target of rapamycin complex (mTOR), further facilitating the phosphorylation of p70 ribosomal S6 kinase 1 (S6K1) and activation of the ribosomal protein S6 (rpS6), which promotes protein synthesis and plays an important role in insulin resistance [19, 20].

In this study, we demonstrate systemic and tissue-specific differences in glucose regulation, baseline fasted insulin levels and hepatic insulin signalling between five C57BL/6 substrains of both sexes, obtained from different commercial vendors. Our data imply that multiple factors affect the metabolic phenotypes observed and stress the importance of selecting and characterising appropriate controls including description of their background, sex, and supplier, as these can be confounders for several fields of biomedical research.

## Materials and methods

### Ethics and experimental rigour statement

This research was prospectively approved by the School of Medicine, Medical Sciences and Nutrition Ethics Review Board (SERB) at the University of Aberdeen. All procedures involving animals were carried out under valid UK Home Office Project Licenses (PPL) with study plans approved by the Medical Research Facility at the University of Aberdeen (Study plan (approval) no: 150621AS (PPL no: P1ECEB2B6), GR_2021_001_CL_olf (PPL no: PP2213334). All research involving animals complied with the EU directive 63/2010E and the United Kingdom (UK) Animals (Scientific Procedures) Act 1986. At the end of the experiment, after a 5-hour fast, animals were humanely culled using cervical dislocation of the neck (Schedule 1 (c) of the Animals (Scientific Procedures) Act 1986).

This work was conducted according to the ARRIVE guidelines 2.0 [21] and the EQIPD framework for rigour in the design, conduct, and analysis of experiments involving animals [22].

### Animals

4-month-old male and female C57BL/6 mice belonging to two different substrains (broadly 6J or 6N) were obtained from two different suppliers: Charles River (Tranent, UK) and Envigo (Netherlands). C57BL6/JCrl (6JCrl) and C57BL6/NCrl (6NCrl) were supplied by Charles River while two different 6J substrains—C57BL6/JOlaHsd (6JOlaEnv), C57BL6/JRccHsd (6JRccEnv), and a 6N substrain C57BL6/NHsd (6NEnv) were supplied by Envigo. A total of 60 animals (Females: n = 4 for 6JCrl, 6NCrl, 6NEnv, n = 4 for 6JOlaEnv, n = 10 for 6JRccEnv; males: n = 6 for 6JCrl, 6NCrl, 6JOlaEnv, 6NEnv, n = 10 for 6JRccEnv) were utilised. All animals were group-housed and habituated in stock cages at the Medical Research Facility (University of Aberdeen) for at least two weeks prior to experimentation. They were maintained on a 12-h day-night cycle (lights on 7am; simulated dusk/dawn 30 min) in temperature and humidity- controlled holding facilities with *ad libitum* access to standard chow (Special Diet Services, Witham, UK), water and enrichment with tail-handling to reduce anxiety exactly as described previously [23]. Genotyping for *Nnt* knockout or *Nnt* wildtype mutation in all substrains was carried out by Transnetyx USA and confirmed that only 6JCrl group had the *Nnt* knockout mutation.

## Glucose Tolerance Test (GTT)

Mice were fasted for 5 hours before GTT between 08:00–13:00 on weekdays in a separate testing room with *ad libitum* access to water maintained as described previously [24]. Fasting blood glucose was measured by a single tail-puncture (time 0) prior to intraperitoneal injection of glucose solution (2mg/g body weight) using microfine syringes to reduce animal suffering. Blood glucose was assessed at 15-, 30-, 60-, and 90-minutes post-injection using AlphaTRAKII veterinary glucometer (Berkshire, UK) post-calibration. One female 6JOlaEnv and 6JRccEnv mouse were excluded from analysis as they had unusually low glucose levels or the glucose levels did not come back to baseline at the end of experiment. Total glycaemic excursion as measured as area under curve (AUC) and blood glucose concentration (mmol/L) at time intervals above were analysed.

## Tissue extraction and sample preparation

For tissue extraction, animals were humanely culled after a 5 hour fast. For serum insulin measurements, trunk blood collected was allowed to clot at room temperature (RT) for 15 mins in separator micro-tubes (BD Microtainer, Canada), followed by centrifugation (7,500 rpm/15 min/4˚C) and serum aliquoting to be stored at -80˚C till use. Extracted livers were snap-frozen in liquid nitrogen and stored in –80˚C until use. For western blotting (WB) experiments, liver tissues were homogenised in RIPA buffer (10 mM Tris-HCl, 150 mM NaCl, 0.1% SDS, 1% Triton, 1% sodium deoxycholate, 5 mM EDTA: pH = 7.4) supplemented with protease inhibitors and PhosStop tablets (Roche). The homogenates were centrifuged at 12,000 rpm for 20 min at 4˚C; supernatants were used in WB.

## Serum insulin ELISA

Serum insulin concentrations were determined using ultra-sensitive mouse insulin ELISA kit (CrystalChem, UK), according to the manufacturer's protocol [24].

## Immunoblotting

WB were performed as described previously [23, 25]. Briefly, protein concentration was adjusted (3 μg/ml) following bicinchoninic acid assay (Sigma Aldrich, UK) and lysates [RIPA buffer, 4X lithium dodecyl sulphate (Fisher Scientific, UK), and 15mM dithiothreitol (DTT)] were heated for 10 min at 70˚C, separated for 45 mins at 200V in MOPS buffer in 4–12% Nupage Bis-Tris gels (Invitrogen) and transferred onto 0.45-μm nitrocellulose (Invitrogen) membranes for 1h at 25V at RT. Next, membranes were washed with 0.05% Tween-20 (Sigma) Tris- buffered saline (TBST) and blocked in TBST with either 5% BSA (bovine serum albumin, Sigma Aldrich) for phosphorylated(p) or 5% skimmed milk powder for total(t) proteins. This was followed by incubating overnight at 4˚C with primary antibodies from Cell Signalling Technologies as follows: (p-AKT(Ser473) Rabbit monoclonal antibody (mAb): #4060, dilution: 1:1000; t-AKT (C67E7) Rabbit mAb: #4691, dilution: 1:1000; p-rpS6 (Ser235/236) (D57.2.2E) XP® Rabbit mAb:#4858, dilution 1:1000; t-rpS6 (5G10) Rabbit mAb: #2217, dilution: 1:1000; p-GSK3β (Ser9) (5B3) Rabbit mAb:#9323, dilution: 1:500; t-GSK3β (27C10) Rabbit mAb: #9315, dilution: 1:1000). Secondary antibodies (goat anti-rabbit IgG-HRP-conjugated; Merck Millipore (1:5000) were then added the next day for 1 hr at RT exactly as described previously [23, 24]. To probe t(otal)-proteins after p(hospho)-proteins, membranes were stripped with a mild stripping buffer (59.52 mM glycine, 81 μM SDS, 1% Tween 20; pH 2.2). They were visualised using enhanced chemiluminescent substrate (ECL; 0.015% hydrogen peroxide ($H_2O_2$), 30 μM coumeric acid in 1.25 mM luminol) on an iBright™ FL1000 Imaging System camera

(Invitrogen™, Fisher Scientific) at 16-bit. Total protein Ponceau-S stain was used as loading control [23]. Densitometric analysis was performed as area under the curve (AUC) using ImageJ (Ver. 1.53, NIH, USA) and normalised vs. Ponceau staining (both p- and t-). Additionally, the ratio of p- to t- protein was calculated. Fold increase of the protein of interest was calculated relative to 6JCrl as these are commonly used in biomedical research as controls [23].

### Statistical analysis

Statistical analysis for GTTs, insulin ELISA and WB was performed using one-way Analysis of Variance (or Welch's ANOVA when standard deviation (SDs) between groups was different) followed by Tukey's post-hoc tests in females and males separately (Prism, Version 8.4, GraphPad USA). Body weight comparisons were carried out using a regular two-way ANOVA with substrain and sex as factors. To interrogate sex differences, pre-planned comparison t-tests between male and female animals of the same substrain were performed. Data are presented as scatter plots with means ± SD. For all analyses, alpha was set to 5% and $p < 0.09$ considered as 'marginally significant' as per recent recommendations [26]. It is acknowledged that in the female cohort, low n's may be a contributing factor for the observance of marginal significances rather than outright significant differences.

### Correlational analysis

Correlational analysis using GraphPad Prism was carried out as described previously [23] between AUC during GTT, serum insulin levels and molecular markers of interest in in males of different substrains as this provided coherent groups with the greatest power. All raw data underwent Z-transformation followed by Pearson's correlation testing (r). Correlations are displayed as heat plot matrices (red: negative; blue: positive correlations). Individual correlational coefficients and their p-values up to 0.09 are represented in the Supplementary Material. Significance was set at $p < 0.05$.

## Results

### Differences in glucose tolerance between C57BL/6 substrains in both sexes

Basal glucose levels and total glycaemic excursion via GTTs were assessed in male and female mice from different C57BL/6 substrains. As expected, body weights of the males overall were higher than females (F (1,48) = 213.4, p<0.0001), and no differences in body weight between substrains were observed (F (4,48) = 1.615, p = 0.18) (Fig 1A).

Importantly, significant differences in glucose tolerance were noted in both sexes (Females: F (4,19) = 5.34 p = 0.004; Males: (F (4,29) = 5.10, p = 0.003). In females, 6JCrl were found to be either significantly or marginally significantly *less* glucose tolerant than most other substrains apart from 6NEnv (Fig 1B) as displayed by higher area under curve during the GTT (6JCrl vs 6JOlaEnv: p = 0.07; 6JCrl vs 6JRccEnv: p = 0.02; 6JCrl vs 6NCrl: p = 0.08). Meanwhile, 6NEnv showed higher total glycaemic excursion (indicating *less* glucose tolerance) compared to 6JRccEnv (p = 0.02) and 6JOlaEnv (p = 0.08).

In male animals (Fig 1C), it was the 6JRccEnv line which was *more* glucose intolerant compared to the 6JOlaEnv (p = 0.05) and 6NCrl (p = 0.002). Additionally, the 6JCrl mice had higher total glycaemic excursion vs. 6NCrl (p = 0.04). A planned paired comparison t-test between male and female 6JRccEnv animals confirmed that male 6JRccEnv animals were much more glucose intolerant compared to the females (p = 0.0004).

Further, the basal glucose levels after 5h of fasting were only different between female substrains (Fig 2A; Welch's ANOVA: F (4.000, 7.851) = 4.013, p = 0.04) but not in males (Fig 2B;

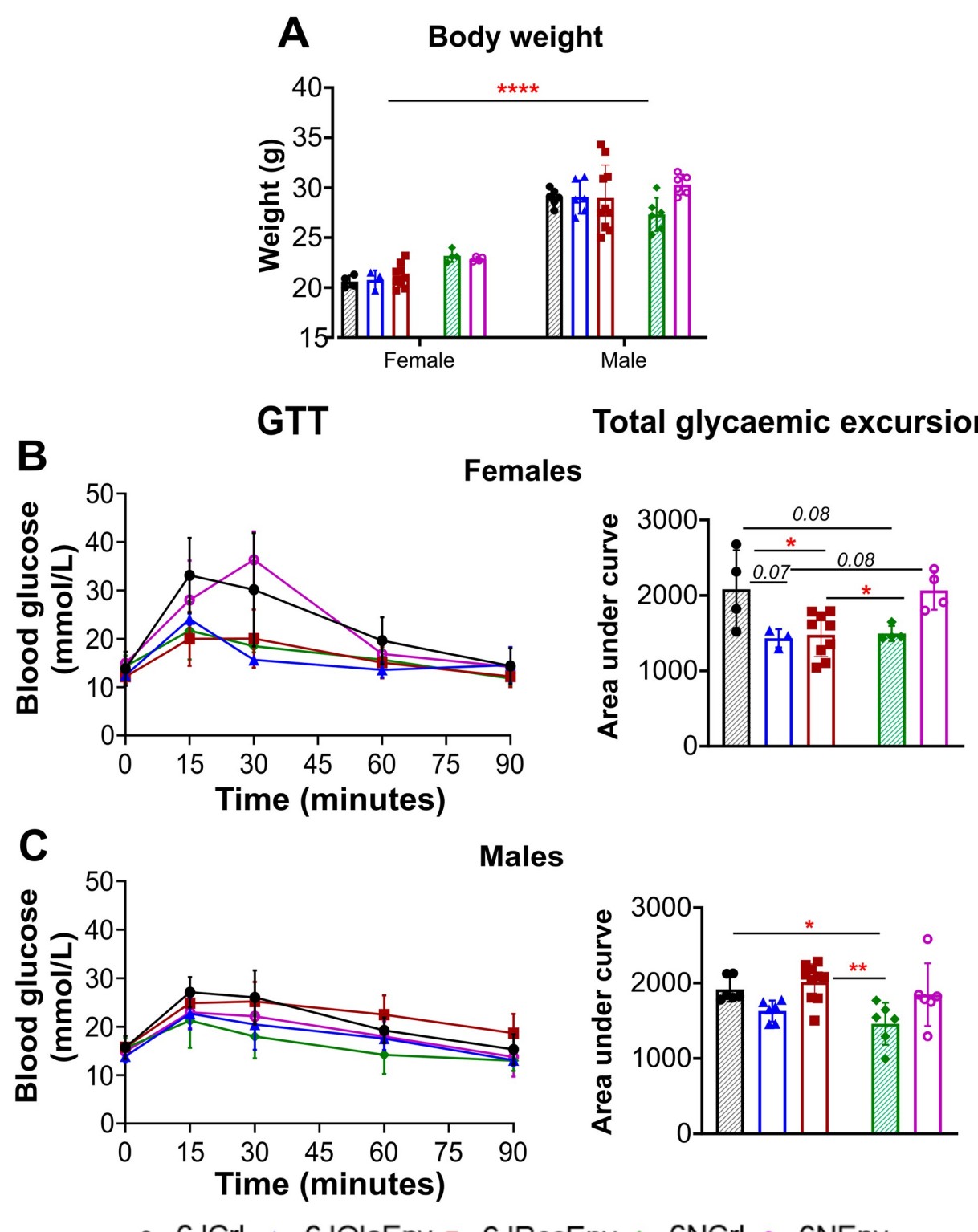

**Fig 1. Body weight and GTT profiles of C57 substrains of both sexes.** Body weight (**A**) of male and female animals belonging to five C57 substrains i.e., 6JCrl (black with shaded pattern), 6JOlaEnv (blue), 6JRccEnv (maroon), 6NCrl (green with shaded pattern), 6NEnv (purple). Blood glucose levels (mmol/L) measured every 15 minutes (left) and total glycaemic excursion measured as total area under curve (right) during the GTT for females (**B**) and male (**C**) C57 substrains. All data sets visualised as scatter plots with mean ± SD with significances for displayed on graph as: ** = p<0.01, * = p<0.05 (Tukey's post-hoc test), p<0.09 in italics for marginal significances (see text for statistics). Key for substrains displayed at bottom of figure with Crl animals (J/N) having shaded pattern to distinguish from Env.

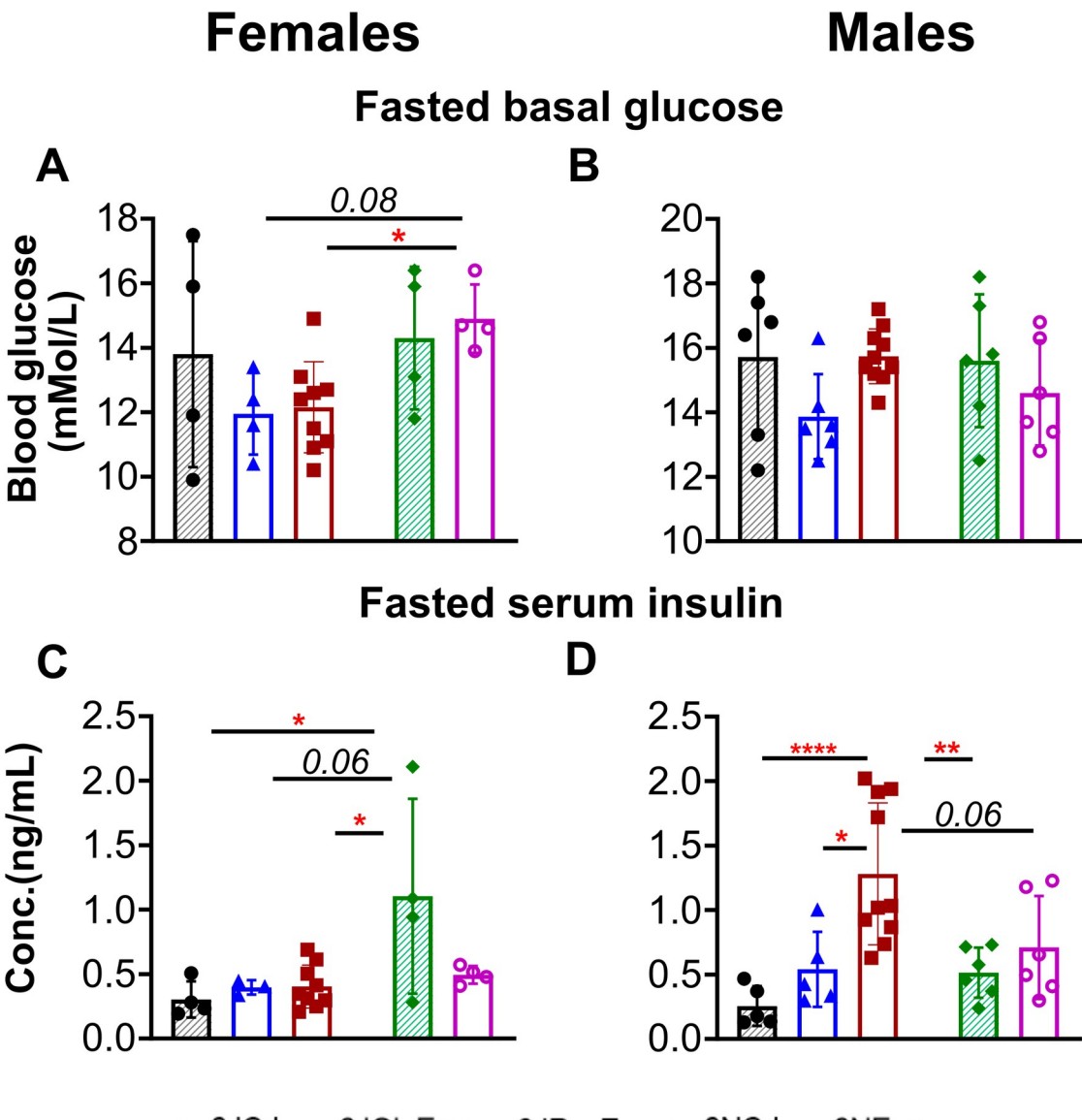

**Fig 2. Basal fasted glucose and serum insulin levels in C57 substrains of both sexes.** Basal glucose and serum insulin levels after a 5hr fast in female (**A, C** respectively) and male (**B, D** respectively) animals belonging to five C57 substrains as described above. All data sets visualised as scatter plots (mean ± SD); significances: **** = p<0.0001, ** = p<0.01, * = p<0.05 (Tukey's post-hoc test), p<0.09 in italics for marginal significances (see main text for statistics). Key to substrains: bottom of figure with Crl animals (J/N) having shaded pattern to distinguish from Env.

F (4,29) = 1.68, p = 0.14). For females, 6NEnv had higher basal fasted glucose levels compared to 6JRccEnv (p = 0.03) and 6JOlaEnv (p = 0.08, marginally significant).

## Strain and sex differences in fasting serum insulin levels

In female mice, 6NCrl female mice exhibited higher serum insulin levels (F (4,19) = 4.071) compared to 6JRccEnv (p = 0.01), 6JOlaEnv (p = 0.06) and 6JCrl (p = 0.01), but not 6Nenv (Fig 2C). A different profile was observed in male animals, where 6JRccEnv displayed the highest serum insulin levels after 5 hours of fasting (F (4,40) = 8.099) compared to 6JOlaEnv (p = 0.03), 6JCrl (p<0.0001), and 6NCrl (p = 0.002) (Fig 2D).

### Strain and sex differences in hepatic insulin signalling

Western blot analysis of the AKT pathway revealed differences between substrains only in male animals (Fig 3F; females are shown in Fig 3B) for p-AKT (F (4,9) = 1.15, p = 0.39), t-AKT (F (4,10) = 1.690, p = 0.22) and the ratio of p-AKT/ t-AKT (p/t-AKT; F (4,10) = 1.909, p = 0.18).

In male animals (Fig 3F), levels of p-AKT in the 6JRccEnv strain were significantly higher than 6NCrl (p = 0.04) and marginally higher (p = 0.08) than 6JCrl mice. No differences in the levels of t-AKT were observed (F<1). Mirroring the pattern observed in p-AKT levels, 6JRccEnv animals had higher p/t-AKT levels compared to both 6JCrl (p = 0.03) and 6NCrl (p = 0.03).

Differences were also observed in the levels of p/t-rpS6 in female animals (Fig 3C), such that 6JCrl mice had higher levels compared to most other substrains (vs 6NCrl, p = 0.01; vs 6JOlaEnv, p = 0.009; vs 6JRccEnv, p = 0.08). No significant differences were observed in either p-rpS6 (F (4,10) = 0.612, p = 0.66) or t-rpS6 (F (4,10) = 0.653, p = 0.63).

Unlike female animals, males (Fig 3G) displayed differences in the levels of t-rpS6. Here, 6JRccEnv had lower levels compared to 6JolaEnv (p = 0.01) and 6NCrl (p = 0.03). No changes were observed in the levels of either p-rpS6 (F (4.24) = 1.5, p = 0.23) or p/t-rpS6 (Welch's ANOVA, W (4,10.40) = 2.53, p = 0.10), although there was a high variability for both markers.

While the levels of p-GSK3β (F (4,10) = 1.25, p = 0.34) and t-GSK3β (F (4,10) = 0.46, p = 0.76) did not change in female animals, differences were observed between substrains (p/t-GSK3β, Fig 3D). Here, 6JOlaEnv had higher levels compared to the other 6J substrains (vs 6JRccEnv, p = 0.03; vs 6JCrl, p = 0.05). Meanwhile, 6NEnv animals had higher levels of p/t-GSK3β compared to 6JRccEnv (p = 0.04) and 6JCr (p = 0.07).

## HEPATIC INSULIN SIGNALLING

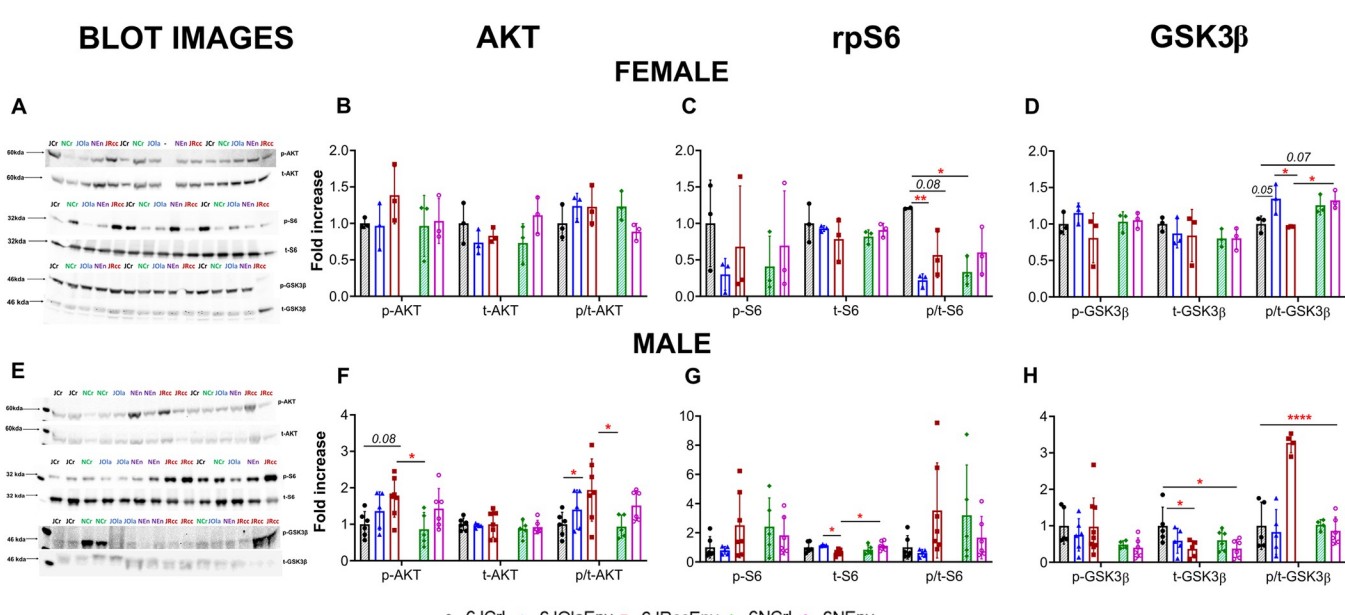

**Fig 3. Hepatic insulin signalling markers in C57 substrains of both sexes.** Representative images of western blots probed with antibodies for the detection of phosphorylated (p-) or total (t-) AKT, rpS6 and GSK3β in liver of female (**A**) and male (**D**) C57 substrains with molecular weight in kDA. Fold increase in protein levels of p-, t-, p/t ratio of: AKT in female (**B**) and male (**F**); rpS6 in female (**C**) and male (**G**); GSK3β in female (**D**) and male (**H**) C57 substrains, relative to 6JCrl group within each sex displayed. All data sets visualised as scatter plots with mean ± SD; significances: **** = p<0.0001 vs all other groups, ** = p<0.01, * = p<0.05 (Tukey's post-hoc test), p<0.09 in italics for marginal significances. Key:6JCrl (black), 6JOlaEnv (blue), 6JRccEnv (maroon), 6NCrl (green), 6NEnv (purple) with Crl animals having shaded pattern inside to distinguish from Env.

In the male cohort (Fig 3H), no differences between substrains were observed for p-GSK3β ($F_{(4, 27)} = 0.94$, $p = 0.45$), but 6JRccEnv ($p = 0.04$) and 6NEnv ($p = 0.03$) animals had lower levels of t-GSK3β compared to 6JCrl males. 6JRccEnv had much higher levels of p/t GSK3β compared to all other substrains (vs 6JCrl, 6NCrl, 6JOlaEnv, 6NEnv, $p<0.0001$). A preplanned comparison between male and female 6JRccEnv for p/t-GSK3β confirmed higher levels in males ($p = 0.0004$).

## Correlational analysis

To interrogate associations between glycaemic excursion during GTTs (i.e. AUC), serum insulin levels and molecular markers (following the format in [23]), we conducted a multi-factorial correlation analysis in 6J male animals (Fig 4). Correlations for 6JCrl, 6JOlaEnv and 6JRccEnv are presented in the main manuscript (see S2 Fig for 6NCrl and 6NEnv correlations in the male animals). We focused on emergence of clusters comparing different insulin signalling markers with (i) AUC during GTT (ii) serum insulin (iii) between themselves and how correlation patterns varied between substrains, rather than searching for individual significances. However, all values have been provided in the Supplement (S1 Fig).

An interesting pattern emerged when comparing GTT AUC values with other markers. Here, a moderate *negative* correlative cluster (red) was observed between AUC and insulin as well as most other hepatic signalling markers only in the 6JCrl animals. The modest *negative* correlation between insulin and AUC in the 6JCrl male animals, although not significant, is interesting as these animals were found to have high glycaemic excursion with low serum insulin levels. Conversely, in the 6JRccEnv animals a weaker, a *positive* (blue) cluster was observed instead. The 6JOlaEnv animals meanwhile had an 'intermediate' phenotype, with AUC positively correlating with insulin levels and GSK3β markers but negatively with other hepatic markers.

## CORRELATIONAL ANALYSIS

**Fig 4. Correlation matrices between GTT glycemic excursion, serum insulin levels and insulin signalling markers in male 6J substrains.** Heat plots depict Z-score converted (overall mean) correlation matrices with positive (blue) and negative (red) correlations based on Pearson's r correlational coefficient, for male 6JCrl (**A**) 6JOlaEnv (**B**), 6JRccEnv (**C**) animals. Upper and lower triangle represent mirror images with GTT indicating area under curve or total glycemic excursion during GTT, insulin indicating basal fasted serum insulin levels followed by hepatic insulin signalling markers as mentioned above. For further detail, see Results.

Serum insulin levels were found to be highly *positively* (and significantly) correlated to most hepatic insulin signalling markers in the 6JCrl animals and to a *much lesser* extent for the 6JRccEnv animals. Insulin levels in the 6JOlaEnv animals had negative correlations with all hepatic signalling markers besides rpS6. This observation makes sense as male 6JOlaEnv animals had both low serum insulin and rpS6 levels but comparatively higher levels of AKT and GSK3β pathway markers.

Correlations between different insulin signalling markers like AKT, rpS6 and GSK3β displayed much stronger positive clusters for the 6JCrl animals compared to either 6JRccEnv or 6OlaEnv mice, where a loss of these correlations was observed. It was also interesting that t-GSK3β correlated *negatively* with serum insulin and other insulin signalling markers but very *positively* with AUC levels (r = 0.87), especially in the 6JCrl animals (and to a lesser extent in the 6JOlaEnv cohort). This is also reflected in the male 6JCrl metabolic profile which showed high levels of t-GSK3β with low levels of insulin and other markers (AKT, GSK3β) as well as high AUC during GTT.

## Discussion

Our findings demonstrate that the metabolic profile and glucose clearance of different C57BL/6 substrains are dependent on sex, substrain and supplier, and thus not exclusively influenced by the presence (or absence) of the *Nnt* mutation.

6JCrl and 6NEnv of both sexes and male 6JRccEnv mice displayed the highest impairments in glycaemic excursion. For animals supplied by Charles River, 6J was marginally more affected compared to 6N with no differences between sexes. For Envigo animals, glucose intolerance profiles of the substrains differed between sexes (6NEnv>6JRccEnv/6JOlaEnv: females; 6NEnv/6JRccEnv>6OlaEnv: males), sometimes even within the same substrain (6JRccEnv males > 6JRccEnv females). In male Envigo mice, despite 100% genetic concordance between the 6JRccEnv and 6JOlaEnv as per Envigo's technical data sheet, the former was found to be more glucose intolerant compared to the latter [17, 18, 27].

Differences between substrains, sexes and vendors may explain why previous publications have reported inconsistent results with regards to GTTs of 6J and 6N animals, further complicated by the omission of their sex in some studies [12–14, 28]. Only a few studies have compared GTT responses on a standard chow diet, as was the case here. It is also interesting to note that 6N substrains were not different from each other, irrespective of vendor or sex, confirming previous results [17].

Based on our observations regarding fasted serum insulin levels, two distinct metabolic phenotypes emerged which could explain impairments in GTTs–one, where low basal insulin levels resulted in impaired glucose clearance during a GTT as seen in 6JCrl of both sexes of our study. This is similar to prior studies in 6J animals (Jackson Laboratory, sex missing) which have impaired glucose tolerance with normal insulin sensitivity but decreased insulin secretion, including a resistance to elevations in circulating insulin after a short-term HFD [5, 12]. The other metabolic phenotype observed in male 6JRccEnv indicated that sustained elevations in fasted basal insulin levels resulted in glucose intolerance [29].

Previous research implicated an *Nnt* mutation in the 6JCrl mice in their impaired glucose clearance. It is clear from the results presented here that *Nnt* deletion is not the only factor responsible for metabolic alterations as substrains with intact *Nnt* (Table 1) also showed glucose intolerance or high insulin levels. Therefore, the *Nnt* mutation at best moderately contributes to disturbed glucose handling [4, 15, 27].

Instead, basal fasted hepatic insulin signalling differed between substrains in both sexes- a likely contributor to variations in metabolic phenotypes observed. For example, p/t-AKT ratio

was greater in male 6JRccEnv animals compared to Crl animals (6J/6N). Increased liver phosphorylation of AKT has previously been associated with higher fasted plasma insulin levels and hyperinsulinemia in C57BL6 animals on a HFD, although no information about substrain, vendor or sex was provided [30]. Like AKT, an increase in p/t-GSK3β levels was also noted in the 6JRccEnv males, where a sex difference was also noted. Incidentally, sex differences in hepatic ischemia/reperfusion injury have been found to be mediated by a male specific gene, SRY (sex-determining-region on the Y chromosome), through the upregulation of GSK3β phosphorylation [31]. Increased GSK3β activation has also been linked to glucose intolerance and insulin resistance, which may explain phenotypes observed in male 6JRccEnv animals [20, 32, 33].

In case of rpS6, substrain differences in p/t-rpS6 were observed only in female animals where 6JCrl presented with the highest ratio. This sex-specific increase in phosphorylation, possibly driven by increased S6K1 activity, could indicate a compensatory mechanism in response to low fasting insulin levels and impaired glucose clearance to initiate further protein translation. In fact, S6K1 deficient mice have been found to be hypo-insulinemic and glucose intolerant, yet are insulin-sensitive and protected against HFD-induced obesity and hepatic steatosis [19, 34, 35].

Finally, results from the multi-factorial correlational analysis in male 6JCrl, 6JOlaEnv and 6JRccEnv animals provide further evidence of substantial differences in metabolic profiles. Here, 6JCrl animals exhibited much stronger and even opposing correlations (e.g., GTT-AUC) compared to 6JRccEnv animals for glucose homeostasis, fasting insulin levels and insulin signalling markers, with 6JOlaEnv animals exhibiting an 'intermediate' phenotype with some similarities to both 6JCrl and 6JRccEnv.

Taken together, 6J and 6N substrains from the same vendor were found to exhibit different metabolic profiles, in accordance with previous literature. These differences were stronger for animals from Envigo compared to Charles River, as we report sex differences even within a particular substrain. The profile of 6JCrl mice of both sexes may be better suited for metabolic studies as they are glucose intolerant, with low fasting insulin levels. In comparison, the 6N substrains tested do not exhibit considerable difference between them or between sexes, irrespective of vendors. Specifically, 6NCrl animals of either sex did not present with basal metabolic phenotypes, which would make them a good choice as healthy controls for a variety of biomedical experiments.

6JRccEnv mice, especially males, presented with glucose intolerance, elevated plasma insulin levels and phosphorylation of insulin signalling markers. Therefore, diabetes or obesity research designed using these mice must bear in mind that their basal metabolic profile is different from other C57BL/6 substrains. The large increase in basal fasted phosphorylation of GSK3β could also have important implications for dementia research since GSK3 overactivity is implicated in the pathogenesis of tau [36].

Finally, our results call for great caution and sufficient inclusion of control experiments for any research where metabolic function may contribute to outcomes. A full characterisation is needed to understand implications on long-term interventions or creation of new transgenic animals. Additionally, sex differences in metabolic profiles indicate the need to consider sex as a biological variable more strongly in metabolic research, as most studies use only male animals [37]. Additionally, while not investigated in the current study, differences in gut microbiota of individual strains has been implicated in metabolic alterations between substrains and should be considered for future studies [38]. Therefore, in line with recent initiatives [1, 21, 22] we strongly recommend that researchers fully and transparently report the sex, substrain and supplier of experimental animals, and consider the use of littermates raised under matching conditions to improve credibility and reproducibility of biomedical research.

## Supporting information

**S1 Fig. Correlation matrices between GTT glycemic excursion, serum insulin levels and insulin signalling markers with values of correlational coefficients (r) and p-values indicated.** Heat plots depict Z-score converted (overall mean) correlation matrices with values of the Pearson's correlational coefficients (positive (blue) and negative (red)) indicated within each of the matrices on top for 6JCrl (**A**) 6JOlaEnv (**B**) 6JRccEnv (**C**) male animals and p-values on the bottom ($<0.09$) for 6JCrl (**D**) 6JOlaEnv (**E**) 6JRccEnv (**F**), respectively. Upper and lower triangle represent mirror images with GTT indicating area under curve or total glycemic excursion during GTT, insulin indicating basal fasted serum insulin levels followed by hepatic insulin signalling markers as mentioned above. For further detail, see Results.
(PDF)

**S2 Fig. Correlation matrices between GTT glycemic excursion, serum insulin levels and insulin signalling markers in male 6N substrains.** Heat plots depict Z-score converted (overall mean) correlation matrices with positive (blue) and negative (red) correlations based on Pearson's r correlational coefficient for male 6NCrl (**A**) and 6NEnv (**B**) animals. Upper and lower triangle represent mirror images with GTT indicating area under curve or total glycemic excursion during GTT, insulin indicating basal fasted serum insulin levels followed by hepatic insulin signalling markers as mentioned above. For further detail, see Results.
(PDF)

**S1 Table. Raw Excel data for the GTT for all groups.**
(XLSX)

**S2 Table. Raw Excel data for serum insulin ELISA for all groups.**
(XLSX)

**S1 File. Raw images of blots and total protein stains (Ponceau) for all groups.**
(PDF)

**S1 Raw images.**
(PDF)

**S1 Data.**
(XLSX)

**S2 Data.**
(XLSX)

## Acknowledgments

We acknowledge staff of the Medical Research Facility for their support with animal care, handling and GTT experiments and members of the Riedel/Platt lab for their assistance with tissue collection and processing. We would also like to acknowledge Prof Gernot Riedel for providing valuable comments and suggestions to this manuscript.

## Author Contributions

**Conceptualization:** Annesha Sil, Marina Souza Matos, Mirela Delibegovic, Bettina Platt.

**Data curation:** Annesha Sil, Marina Souza Matos.

**Formal analysis:** Annesha Sil, Marina Souza Matos.

**Funding acquisition:** Mirela Delibegovic, Bettina Platt.

**Investigation:** Annesha Sil, Marina Souza Matos, Bettina Platt.

**Methodology:** Annesha Sil, Marina Souza Matos.

**Project administration:** Bettina Platt.

**Resources:** Mirela Delibegovic, Bettina Platt.

**Software:** Annesha Sil, Marina Souza Matos.

**Supervision:** Mirela Delibegovic, Bettina Platt.

**Validation:** Annesha Sil, Marina Souza Matos, Mirela Delibegovic, Bettina Platt.

**Visualization:** Annesha Sil, Marina Souza Matos, Bettina Platt.

**Writing – original draft:** Annesha Sil, Marina Souza Matos, Mirela Delibegovic, Bettina Platt.

**Writing – review & editing:** Annesha Sil, Marina Souza Matos, Mirela Delibegovic, Bettina Platt.

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
