## [Decision Letter · Decision Letter 0]

7 Jun 2023

PONE-D-23-11737How stra(i)nge are your controls?  A comparative analysis of metabolic phenotypes in commonly used C57 substrains

PLOS ONE

Dear Dr. Sil,

Thank you for submitting your manuscript to PLOS ONE. After careful consideration, we feel that it has merit but does not fully meet PLOS ONE’s publication criteria as it currently stands. Therefore, we invite you to submit a revised version of the manuscript that addresses the points raised during the review process.

We look forward to receiving your revised manuscript.

Kind regards,

Salvatore V Pizzo

Academic Editor

PLOS ONE

Journal Requirements:

Additional Editor Comments:

Both reviewers have raised issues which while minor should be considered by the authors.

Reviewers' comments:

Reviewer's Responses to Questions

**Comments to the Author**

1. Is the manuscript technically sound, and do the data support the conclusions?

Reviewer #1: Yes

Reviewer #2: Yes

2. Has the statistical analysis been performed appropriately and rigorously? 

Reviewer #1: Yes

Reviewer #2: Yes

3. Have the authors made all data underlying the findings in their manuscript fully available?

Reviewer #1: Yes

Reviewer #2: Yes

4. Is the manuscript presented in an intelligible fashion and written in standard English?

Reviewer #1: Yes

Reviewer #2: Yes

5. Review Comments to the Author

Reviewer #1: This study by Sil A et al addresses an important aspect of animal research related to phenotypic differences between mouse strains and substrains, an aspect that is still not considered and reported sufficiently in research. This has and continue to contribute to the reproducibility issues in science. As such, it is critical to highlight and report differences between mouse models to raise awareness and strenghten the message to the scientific community. The work by Sil et al is thus timely and brings novel insights on metabolic differences between substrains, sex and vendors that are only partly explained by the Nnt mutation status. This reinforces the idea that the previously reported difference in GSIS in 6J and 6N is not fully explained by the Nnt mutation. In addition, they provide interesting data showing that there are differences in the hepatic insulin signaling pathway in part correlated with insulin levels and/or GTT AUC. Overall, the experimental design is strong and the studies well conducted. The article is really clear and well written. The results are strong and convincing. Animal number in some female groups is on the low side but this limitation is acknowledged in the methods and considered for the statistical analysis and data interpretation. I have only minor comments:

1-Basal glycemia and insulin levels could be presented in Fig 1 with BW data while GTT and AUC data presented as Fig 2. It sounds more logical to this reviewer at least.

2-There are evidence in the literature that microbiota could be involved in metabolic differences between mouse strains and substrains. See for instance PMID: 26299453. This aspect should be discussed also as a potential contributor to the differenecs observed in the current study.

Reviewer #2: I have a few minor points of criticisms to the manuscript by Sil and collaborators.

- Manuscript title: “C57BL/6 substrains” should be more appropriate than “C57 substrains”.

- Manuscript text: please replace “sub-strains” with “substrains”.

- Define the Nnt mutation as Nnt^C57BL/6J when it first appears (on line 29). See https://www.informatics.jax.org/allele/MGI:3626282

- Line 60: Remove “(NAD)”

- Line 61: Please remove “, required for ATP synthesis”.

- Line 61: NADPH instead of NAPDH.

- Line 61: “NADPH is required for biosynthesis and redox homeostasis” instead of “NAPDH detoxifies reactive oxygen species (ROS) in mitochondria, essential for H2O2 disposal and redox homeostasis”.

- Results section: Remove figure captions (Figures 1-4) from the text.

- I suggest authors cite a critical review on mouse background strain and the history of the Nnt mutation in C57BL/6J:

Diabetes. 2016 Jan;65(1):25-33. doi: 10.2337/db15-0982.

- I suggest that the authors cite a critical review on the effects of the Nnt mutation on mouse obesity and several other parameters:

Antioxid Redox Signal. 2022 May;36(13-15):864-884. doi: 10.1089/ars.2021.0111.

6. PLOS authors have the option to publish the peer review history of their article (what does this mean?). If published, this will include your full peer review and any attached files.

Reviewer #1: No

Reviewer #2: No

---

## [Author Response · Author response to Decision Letter 0]

28 Jun 2023

Response to the reviewers

Manuscript ID: PONE-D-23-11737R1 

‘How stra(i)nge are your controls? A comparative analysis of metabolic phenotypes in commonly used C57BL/6 substrains’

Dear Prof. Pizzo,

Thank you very much for considering our manuscript for publication in your journal, and for the encouraging news on our submission. 

We are grateful to the reviewers for their valuable comments and appreciations. All comments have been carefully considered, and the manuscript has been amended according to the suggestions made. Please find a point-by-point overview attached. 

The minimum underlying data set which includes raw Excel data files for the Glucose Tolerance Test, serum insulin ELISA, raw images of both total protein Stain Ponceau and raw images of the blots have now been uploaded as Supporting Information files to the manuscript. 

As requested by the reviewers, two new references have been added to the reference list (PMID: 26299453, doi: 10.1089/ars.2021.0111). The reference list has also been updated as requested. 

We hope the manuscript is now suitable for publication, but please do not hesitate to contact us if there are any further questions.

We would like to thank you for the speedy processing of our submission and trust that PLOS One readers will find our contribution interesting. 

Sincerely,

Annesha Sil and Bettina Platt

 

Reviewer #1: 

We are very pleased that the reviewer considers this to be a well-conducted and strong study and one which addresses an important aspect of animal research. 

Replies to specific points follows: 

1-Basal glycemia and insulin levels could be presented in Fig 1 with BW data while GTT and AUC data presented as Fig 2. It sounds more logical to this reviewer at least.

Reply: We thank the reviewer for this comment. We would, however, prefer to maintain the figures as they currently are. For us, maintaining Fig.1 with body weight and glucose tolerance and AUC seems more logical as these were evaluated earlier in the experimental timeline, whereas serum insulin was evaluated post-mortem. We do understand that basal glucose was also evaluated during the glucose tolerance test however we did not want to increase the number of figures for the manuscript to keep it as succinct as possible and hence this has been combined with the fasted insulin levels in Fig. 2. 

2-There are evidence in the literature that microbiota could be involved in metabolic differences between mouse strains and substrains. See for instance PMID: 26299453. This aspect should be discussed also as a potential contributor to the differences observed in the current study.

Reply: Indeed, the reviewer raises an important point. Accordingly, an additional line has been added in the discussion from line 363-365 with the reference the reviewer has mentioned as follows: ‘Additionally, while not investigated in the current study, differences in gut microbiota of individual strains has been implicated in metabolic alterations between substrains and should be considered for future studies [38].’

Reviewer #2: 

We thank the reviewer for their valuable suggestions. 

Replies to specific points follows: 

- Manuscript title: “C57BL/6 substrains” should be more appropriate than “C57 substrains”.

Reply: This has now been amended.

- Manuscript text: please replace “sub-strains” with “substrains”.

Reply: This has now been altered as requested.

- Define the Nnt mutation as Nnt^C57BL/6J when it first appears (on line 29). See https://www.informatics.jax.org/allele/MGI:3626282

Reply: This has now been changed. 

- Line 60: Remove “(NAD)” 

Reply: This has now been changed. 

- Line 61: Please remove “, required for ATP synthesis”.

- Line 61: NADPH instead of NAPDH.

- Line 61: “NADPH is required for biosynthesis and redox homeostasis” instead of “NAPDH detoxifies reactive oxygen species (ROS) in mitochondria, essential for H2O2 disposal and redox homeostasis”.

Reply: This line has been changed as requested.

- Results section: Remove figure captions (Figures 1-4) from the text.

Reply: We apologise, however this cannot be altered as we were just following the journals submission guidelines which request the caption to be placed right after the paragraph in which it first appears

- I suggest authors cite a critical review on mouse background strain and the history of the Nnt mutation in C57BL/6J:

Diabetes. 2016 Jan;65(1):25-33. doi: 10.2337/db15-0982.

Reply: This critical review was already in the reference list (Reference no. 2, line 46). We apologise if this was unclear in any form.

- I suggest that the authors cite a critical review on the effects of the Nnt mutation on mouse obesity and several other parameters:

Antioxid Redox Signal. 2022 May;36(13-15):864-884. doi: 10.1089/ars.2021.0111.

Reply: We thank the reviewer for this suggestion. This has now been added to line 64 as an additional reference number [11].

---

## [Decision Letter · Decision Letter 1]

19 Jul 2023

How stra(i)nge are your controls?  A comparative analysis of metabolic phenotypes in commonly used C57BL/6 substrains

PONE-D-23-11737R1

Dear Dr. Sil,

We’re pleased to inform you that your manuscript has been judged scientifically suitable for publication and will be formally accepted for publication once it meets all outstanding technical requirements.

Kind regards,

Salvatore V Pizzo

Academic Editor

PLOS ONE

Additional Editor Comments (optional):

Reviewers' comments:

Reviewer's Responses to Questions

**Comments to the Author**

1. If the authors have adequately addressed your comments raised in a previous round of review and you feel that this manuscript is now acceptable for publication, you may indicate that here to bypass the “Comments to the Author” section, enter your conflict of interest statement in the “Confidential to Editor” section, and submit your "Accept" recommendation.

Reviewer #1: All comments have been addressed

Reviewer #2: All comments have been addressed

2. Is the manuscript technically sound, and do the data support the conclusions?

Reviewer #1: Yes

Reviewer #2: Yes

3. Has the statistical analysis been performed appropriately and rigorously? 

Reviewer #1: Yes

Reviewer #2: Yes

4. Have the authors made all data underlying the findings in their manuscript fully available?

Reviewer #1: Yes

Reviewer #2: Yes

5. Is the manuscript presented in an intelligible fashion and written in standard English?

Reviewer #1: Yes

Reviewer #2: Yes

6. Review Comments to the Author

Reviewer #1: (No Response)

Reviewer #2: (No Response)

7. PLOS authors have the option to publish the peer review history of their article (what does this mean?). If published, this will include your full peer review and any attached files.

Reviewer #1: No

Reviewer #2: No

---

## [Editor Report · Acceptance letter]

24 Jul 2023

PONE-D-23-11737R1 

How stra(i)nge are your controls? A comparative analysis of metabolic phenotypes in commonly used C57BL/6 substrains 

Dear Dr. Sil:

I'm pleased to inform you that your manuscript has been deemed suitable for publication in PLOS ONE. Congratulations! Your manuscript is now with our production department. 

Kind regards, 

on behalf of

Dr. Salvatore V Pizzo 

Academic Editor

PLOS ONE